# Analysis of Spatial Relationship Based on Ecosystem Services and Ecological Risk Index in the Counties of Chongqing

**Zihui Li** [1,2,†], **Kangwen Zhu** [3,†], **Dan Song** [4,5], **Dongjie Guan** [3], **Jiameng Cao** [3], **Xiangyuan Su** [4,5], **Yanjun Zhang** [6,*], **Ya Zhang** [1,2], **Yong Ba** [1,2] and **Haoyu Wang** [1,2]

1 China Geological Survey Kunming General Survey of Natural Resources Center, Kunming 650100, China; lizihui@mail.cgs.gov.cn (Z.L.); zhangya@mail.cgs.gov.cn (Y.Z.); bayong@mail.cgs.gov.cn (Y.B.); m18813097066@163.com (H.W.)
2 Technology Innovation Center for Natural Ecosystem Carbon Sink, Ministry of Natural Resources Kunming, Kunming 650100, China
3 School of Smart City, Chongqing Jiaotong University, Chongqing 400074, China; zhukangwen@email.swu.edu.cn (K.Z.); guandongjie_2000@163.com (D.G.); cjm_1998225@163.com (J.C.)
4 Chongqing Academy of Ecology and Environmental Sciences, Chongqing 401147, China; causd@126.com (D.S.); suxiangyuan_1997@163.com (X.S.)
5 Southwest Branch of Chinese Academy of Environmental Sciences, Chongqing 401147, China
6 School of Business Management, Chongqing University of Technology, Chongqing 400050, China
* Correspondence: zhangyanjun@cqut.edu.cn
† These authors contributed equally to this work.

**Abstract:** Due to the insufficient research on the spatial relationship and driving mechanism of ecosystem services and ecological risks and the current background of rising ecological risks and dysfunctional ecosystem services in local areas, analyzing the relationship and driving mechanism is an urgent task in order to safeguard regional ecological security and improve ecosystem services at present. Taking Chongqing as an example, the study scientifically identifies the spatial relationship between ecosystem services and ecological risks and their driving factors at district and county scales based on the constructed Ecosystem Service—Driver–Pressures–Status–Impacts–Responses (ES-DPSIR) model. The main findings include (1) significant variation in the spatial distribution of the comprehensive ecosystem service index, where the lowest ecosystem service index (0.013) was found in the main urban area of Chongqing and the scores gradually increased outward from this center, reaching 0.689 in the outermost areas; (2) an increase in the comprehensive ecological risk index from east to west, ranging from −0.134 to 0.333; (3) a prominent spatial relationship between ecosystem services and ecological risks, with 52.63% of the districts and counties being imbalanced or mildly imbalanced; and (4) significant differences between development trends of ecosystem services—ecological risks, including 60.53% imbalanced and 30.47% mildly balanced districts. This study identified and analyzed the spatial change characteristics of ecosystem services and ecological risks based on the ES-DPSIR model, explored the driving factors, and provided new ideas for the relationship and driving research. The results of the study could provide effective ways and references for improving regional ecological security and enhancing the capacity of ecosystem services.

**Keywords:** ecosystem services; ES-DPSIR model; spatial relationships; driving factors

## 1. Introduction

Nowadays, increasing demand for ecosystem services is essential for our daily lives, but rapid economic development has caused a profound impact on the environment [1], resulting in unprecedented changes in the structure and function of these ecosystems. Meanwhile, irresponsible city planning, development, and other human activity have caused many ecological and environmental issues [2,3], leading to increased regional ecological risks [4]. The decline of ecosystem services and the deterioration of regional

environments is extremely unfavorable for the long-term development of humans. Based on this predicament, ensuring regional ecological security has become a key issue for humans. Ecosystem services guarantee sustainable development and provide various services and benefits to humans [5], whereas ecological risks focus on the environmental effects caused by both nature and society [6]. Associations between ecosystem services and ecological risks have been found. Therefore, it is necessary to assess the coordination between ecosystem services and ecological risks and to explore mutual feedback mechanisms from the perspective of the human–land system, which can improve regional ecosystem service capabilities and prevent regional ecological risks, hence promoting the coordinated development of ecosystem services and ecological risks.

Various studies have been conducted on ecosystem services and ecological risks, with ecosystem services and ecological risks usually treated as two separate topics. Studies on ecosystem services have mainly focused on ecosystem service evaluation [7,8], ecosystem service relationship evaluation [9–11], and driving mechanism identification [12,13]. At present, a research paradigm of "function–pattern–scale–relationship–drive" has been formed in studies on ecosystem services. On the other hand, studies on ecological risks have mainly focused on the development of an ecological risk index system [14], model development [15], ecological risk evaluation [16–20], ecological security pattern development [21,22], ecological risk early warning/simulation [23], and ecological risk spatial identification [24]. Overall, studies on ecological risk research have also formed a unique paradigm, one of "pattern–scale–drive–warning–recognition". With the deepening of research, ecological risks and ecosystem services have shown the development characteristics of independent to integrated. The integration of the two can effectively correlate ecological processes and human well-being and has become the research frontier and hotspot of ecological security [25,26]. Research on the integration of ecological risks and ecosystem services has focused on integrating ecosystem services into ecological risk assessment systems [16,27]. Wang et al. [28] and Ouyang et al. [29] conducted an ecological risk assessment based on ecosystem services and ecosystem health, which provides a new perspective for ecological risk management. Xie et al. [30] carried out ecological zoning based on ecosystem services and ecological risk characteristics of ecological functional areas. Proposing management strategies of risk areas based on ecosystem service functions to determine relevant ecological risks has become a new research direction [31,32]. In addition, the correlation between ecosystem services and ecological risks has also been explored, mainly using correlation analysis [33] and grey relationship analysis [34]. Although some studies have explored the relationship between ecosystem services and ecological risks, there is still no research on the degree of impact of ecological risks on ecosystem services and the impact of their functions, so it is difficult to further reveal the essence of the relationship. In view of the lack of research on the spatial relationship and driving mechanism of ecosystem services and ecological risks, coupled with the background of the current rise of ecological risks in local areas and the dysfunction of ecosystem services, the analysis of their relationship and the driving mechanism are key issues that need to be resolved to ensure regional ecological security and improve ecosystem services.

Chongqing is located in Southwest China and has complex topography, landforms, climate, hydrology, and other natural features. It is an important ecological area for the upper reaches of the Yangtze River and is significant to the local ecosystem. With the rapid development of the regional economy, the reduction in ecosystem services and the increase in ecological risks in some areas have led to difficulties in the harmonious coexistence between man and nature, challenging the sustainable development of the regional economy. Especially in recent years, driven by economic development, the environmental development of various districts and counties in Chongqing has gradually highlighted huge differences. From perspective of the natural environment and socio-economic perspectives, the unique geomorphology leads to differences in the spatial distribution of ecosystem service functions, and the varying human social activity disturbances within the region generate different levels of ecological risk. Therefore, it is necessary and valuable to study

the relationships and drivers of ecosystem services and ecological risks at the district and county scales in Chongqing. Based on the current background and situation, we selected Chongqing as the research object to explore the following scientific questions: How can the spatial relationship and evolution trend between ecosystem services and ecological risks at the district and county scales be scientifically identified? What are the drivers of the ecosystem services–ecological risk spatial relationship? Taking Chongqing's county as the research scale, firstly, the ES-DPSIR system was constructed from the two dimensions of ecosystem service and ecological risk; secondly, the spatial relationship and the development trend of the ecosystem service dimension comprehensive index and the ecological risk dimension comprehensive index were measured, respectively; and finally, the driving force was discussed. The innovation of the study mainly lies in the construction of the ES-DPSIR model to analyze the spatial relationship and development trend of ecosystem services and ecological winds, which can provide new ideas for studying the relationship and drivers. The goal is to provide a basis for the optimization and management of the relationship between ecosystem services and ecological risks and reduce ecological risks while improving regional ecosystem services.

## 2. Materials and Methods

### 2.1. Study Area

Chongqing is located in the southwest of the Chinese mainland, the upper reaches of the Yangtze River, and the southeast of the Sichuan Basin, spanning the transition zone between the Qinghai–Tibet Plateau and the middle and lower reaches of the Yangtze River between $105°11'$–$110°11'$ east longitude and $28°10'$–$32°13'$ north latitude (Figure 1). As the only municipality directly under the Central Government in Western China, Chongqing is an important site for China's "One Belt, One Road" strategy. With a land area of 82,400 km², there are important water systems such as the Yangtze River and the Jialing River flowing through the area, providing abundant water and superior natural river resources. By the end of 2020, the population of Chongqing had reached 3.205 million and the GDP had exceeded CNY 25,000 billion (a 3.9% increase from the previous year), with raw materials, manufacturing, and service industries contributing to 7.2%, 40.0%, and 52.8% of the GDP, respectively. With the rapid development of the economy, Chongqing's ecological environment is facing severe problems and challenges. The frequent occurrence of environmental issues (i.e., chemical pollution, soil erosion, landslides, deterioration of ecological environment quality, and imbalance of ecological structures) in some areas has become a bottleneck for coordinated development between industry, the environment, and the economy. It is crucial to understand how to promote the simultaneous improvement of regional ecosystem services and ecological security together with the rapid development of the economy. In this study, Chongqing is divided into several different areas based on economic, social, and topographical characteristics, including the main urban area, Western Chongqing, Southeastern Chongqing, and Northeastern Chongqing. The main urban areas include Yuzhong, Dadukou, Jiangbei, Shapingba, Jiulongpo, Nan'an, Beibei, Yubei, and Banan districts; the western area includes Fuling, Changshou, Jiangjin, Hechuan, Yongchuan, Nanchuan, Qijiang, Dazu, Bishan, Tongliang, Tongnan, and Rongchang; the northeast area includes Wanzhou, Kaizhou, Liangping, Chengkou, Fengdu, Dianjiang, Zhongxian, Yunyang, Fengjie, Wushan, and Wuxi; and the southeast area includes Qianjiang, Wulong, Shizhu, Xiushan, Youyang, and Pengshui.

### 2.2. Data Sources

The data used in the study include land use data, administrative boundary data, soil spatial attribute data, and spatial meteorological interpolation data. The land use data, administrative boundary data, and spatial meteorological data were obtained from the Resource and Environmental Science and Data Center of the Chinese Academy of Sciences (resdc.cn) and had a spatial resolution of 1000 m. The soil spatial attribute data were obtained from the National Glacier Permafrost and Desert Science Data Center (ncdc.ac.cn)

and had a spatial resolution of 1000 m. All of the data were converted to uniform projected coordinates and transformed to a uniform projected coordinate system in ArcGIS. The statistical data, including GDP per capita and disposable income, are from the 2020 Statistical Yearbook of Chongqing counties and districts.

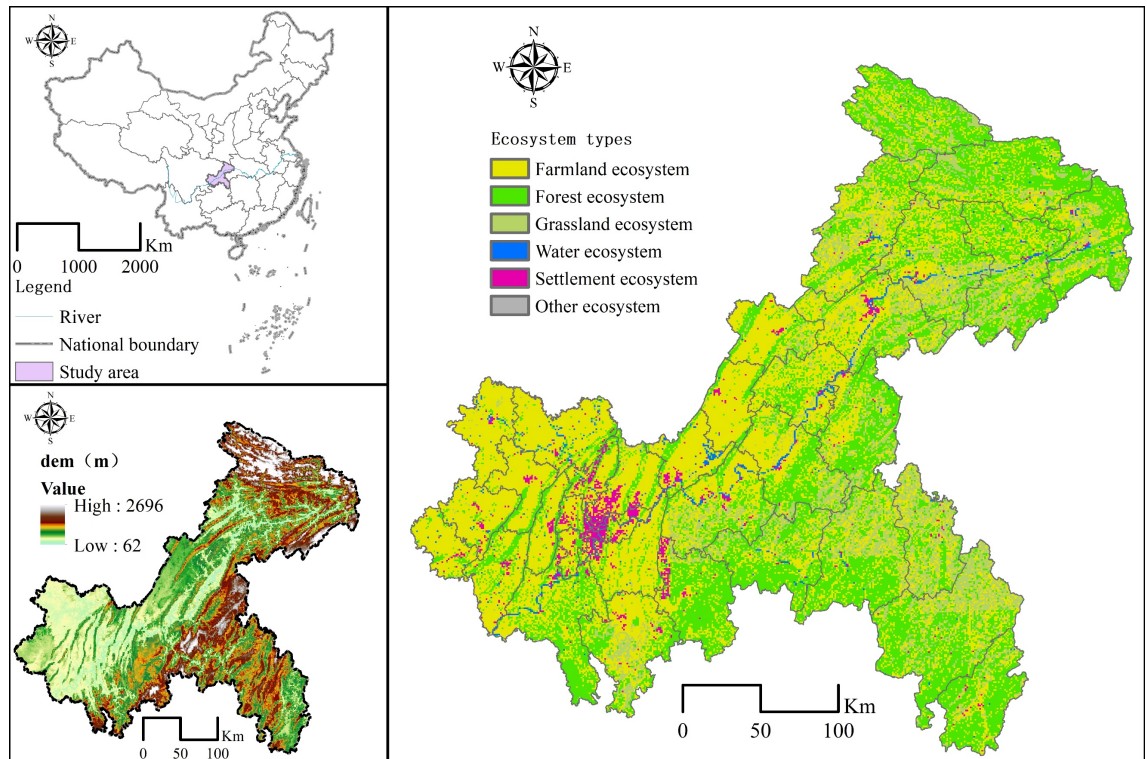

**Figure 1.** Schematic location of the study area.

### 2.3. Methods

#### 2.3.1. Research Framework

Ecosystem services refer to the benefits that humans receive from ecosystems and play an important role in maintaining the stability of the ecosystem environment and sustainable development [35]. Ecological risk refers to the possibility that an ecosystem will be affected by any elements outside of the ecosystem that pose a threat to the ecosystem. The effects of uncertain events or disasters in the region on ecosystems and their components can lead to damage to ecosystem structure and function [36]. From the perspective of the relationship between the two, the decline of ecosystem services will lead to a decline in the stability of the ecosystem and increase the ecological risk [17]. Ecosystem services include supply, regulation, and cultural services, including water conservation, soil conservation, carbon sequestration, biodiversity, etc. Ecological risk is mainly evaluated based on the Driver–Pressures–Status–Impacts–Responses model (DPSIR), which includes natural, ecological, economic, and social impact driving indicators [23].

The overall framework is shown in Figure 2, which includes two dimensions, the ecosystem service index (ESI) and the ecological risk index (ERI). On the one hand, the ecosystem service index was measured in terms of supply, regulation, and spiritual and cultural aspects, specifically including 8 indicators, such as food production, water supply, water conservation, and air purification. On the other hand, the risk index was measured based on the corresponding five dimensions of Driver–Pressures–Status–Impacts–Responses (DPSIR) in the context of nature, the economy, and society, specifically including 18 indicators, such as GDP per capita, population, urbanization level, population growth rate, and so on. Finally, the Ecosystem Service—Driver–Pressures–Status–Impacts–Responses (ES-DPSIR)

model based on double indices (ESI and ERI) was formed to analyze and explore the relationship.

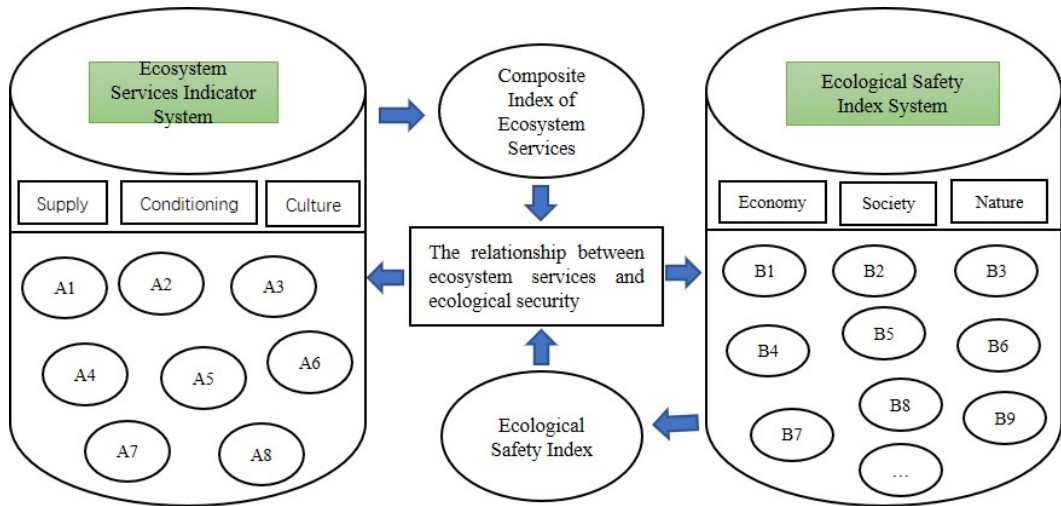

**Figure 2.** The framework of this study.

2.3.2. Calculation of Comprehensive Ecosystem Service Index

(1)    Development of the ecosystem service index system

The comprehensive ESI represents the overall condition of all regional ecosystem services, and thus, the indicators need to be representative and comprehensive [37]. Based on previous studies [38], the indicators that characterize the capacity of ecosystem services were mainly selected from the three dimensions of provisioning, regulating, and culture. In this study, eight representative ecosystem service indicators were selected from the three dimensions of supply, regulation, and culture to calculate the ESI. The weight of each indicator was calculated using the entropy weight method [23]. Details on the ESI system are shown in Table 1.

**Table 1.** The indicators of the ecosystem service index system.

| Index System | Level I Index | Level II Index | Abbr. | Description | Unit | Weight |
|---|---|---|---|---|---|---|
| Ecosystem service index | Supply | Food production | A1 | The ability of ecosystems to provide food | t | 0.0901 |
| | | Water supply | A2 | The ability of ecosystems to provide water | mm | 0.1971 |
| | Regulation | Soil conservation | A3 | The ability of ecosystems to maintain soil | t | 0.0962 |
| | | Air purification | A4 | The ability of ecosystems to purify pollution | t | 0.1110 |
| | | Carbon storage | A5 | The ability of ecosystems to store carbon | t | 0.1129 |
| | | Habitat quality | A6 | The ability of ecosystem to maintain biodiversity | - | 0.1642 |
| | | Climate regulation | A7 | The ability of ecosystems to regulate climate | KWh | 0.0979 |
| | Culture | Leisure | A8 | The ability of ecosystems to provide residents with travel and leisure | 10,000 people | 0.1305 |

(2)    Calculation of the ecosystem service index

In this study, the Level II indicators of the ESI were calculated based on statistical surveys and InVEST models. The detailed methodology for the calculation of each indicator is shown in Table 2.

**Table 2.** Methodology for the calculation of the ecosystem service index.

| Index | Method | Principle and Formula | Description of the Parameters |
|---|---|---|---|
| Food production | Statistical survey | Grain supply data from the Statistical Yearbook | - |
| Water supply | InVEST model [39] | Quantitatively evaluate the water production capacity of the ecosystem based on the principle of water balance $Y = \left(1 - \frac{AEP}{P}\right) \times P$ | Y is the average annual water production, AEP is the annual actual evapotranspiration, and $p$ is the average annual precipitation. |
| Soil conservation | InVEST model [39] | Comprehensively evaluate the ability of the parcel to intercept upstream sediments based on the general soil loss equation $USLE = R \times K \times LS(1 - C \times AP)$ | USLE is the soil conservation per unit area, R is the rainfall erosion factor, K is the soil erodibility factor, LS is the slope length and slope factor, and C and AP are the vegetation cover management factor and the soil conservation measurement factor, respectively. |
| Air purification | Air purification model [40] | Assessed by the ability of vegetation to purify pollutants $Q_{ap} = \sum_{i=1}^{m} \sum_{j=1}^{n} Q_{ij} \times A_i$ | $Q_{ap}$ is the air purification amount and $Q_{ij}$ is the purification amount of vegetation $i$ to the air pollutant $j$ per unit area ($i = 1, 2, \ldots, m$, unitless, and $j = 1, 2, \ldots, n$, unitless). $A_i$ is the area of vegetation $i$, m is the amount of vegetation $i$, and $n$ is the number of air pollutants. |
| Carbon storage | InVEST model [39] | Comprehensively consider the carbon storage of carbon above ground, underground, in soil, and in dead biomass $C = C_{above} + C_{below} + C_{soil} + C_{dead}$ | $C$ is the total carbon storage, $C_{above}$ is the aboveground biological carbon storage, $C_{below}$ is the underground biological carbon storage, $C_{soil}$ is the soil carbon storage, and $C_{dead}$ is the dead organic matter carbon storage. |
| Habitat quality | InVEST model [39] | Habitat quality was assessed using the habitat-quality module in the InVEST model. $Q = H \times \left(1 - \left(D^2 / (D^2 + K^2)\right)\right)$ | $Q$ is the habitat-quality index, $H$ is the habitat suitability of the ecosystem, $D$ is the habitat degradation degree, $K$ is the half-saturation constant (half of the maximum degradation degree was used), and $z$ is a normalized constant. |
| Climate regulation | Climate regulation model [40] | The regulation capacity of vegetation and water were considered for climate regulation. $E_{pt} = \sum_{i}^{3} EPP_i \times S_i \times TD \times 10^6 / (3600 \times r)$ $E_{pt} = E_w \times e \times 10^3 / 3600 + E_w \times y$ | $E_{pt}$ is the energy consumed by the transpiration of farmland vegetation, $E_{PPi}$ is the transpiration of heat consumption per unit area of vegetation $i$, $S_i$ is the area of vegetation $i$, $TD$ is the number of days when the daily maximum temperature is above 26 °C, $E_{we}$ is the total energy consumed by the ecosystem to adjust temperature or humidity, $E_w$ is the amount of evaporation, $e$ is the latent heat of volatilization (the heat required to evaporate 1 g of water), and y is the electricity consumption of the humidifier to convert 1 m³ of water into steam. |
| Leisure | Statistical survey | The number of tourists from the Statistical Yearbook was used. | - |

(3)  Calculation of the comprehensive ecosystem service index

Different ecosystem services indicate the ability of ecosystems to maintain and regulate human well-being. Thus, in order to accurately evaluate the comprehensive ESI, the indicators were normalized prior to further assessment. The weight of each normalized ESI indicator was evaluated using entropy weight [23]. The comprehensive index was then calculated using the comprehensive score method. The formulas for the comprehensive ESI are shown below:

$$ESS_{pq} = \frac{ES_{pq} - min\{ES_{pq}, \ldots, ES_{nq}\}}{max\{ES_{pq}, \ldots, ES_{nq}\} - min\{ES_{pq}, \ldots, ES_{nq}\}} \tag{1}$$

where $ESS_{pq}$ is the normalized ecosystem service $q$ in indicator $p$, max($ES_{pq}, \ldots, ES_{nq}$) is the maximum value of each indicator, and min($ES_{pq}, \ldots, ES_{nq}$) is the minimum value of each indicator.

$$ESI = \sum W_d \times ESS_d \tag{2}$$

where $W_d$ is the weight of each *ESI* indicator, and $ESS_d$ is the normalized value of the ecosystem service.

### 2.3.3. Calculation of Comprehensive Ecological Risk Index

(1)  Development of the ecological risk index system

As an indicator of the potential threats to the ecosystem, the ecological risk is the result of various factors in the ecological environment. Thus, it is important to accurately and comprehensively select the indicators for the ERI system. In this study, an ERI system was developed based on the Driver–Pressures–Status–Impacts–Responses (DPSIR) framework, and the corresponding weight of each index was calculated using the entropy weight (Table 3).

**Table 3.** Driver–Pressures–Status–Impacts–Responses (DPSIR) index system.

| Indicator System | Level I Indices | Level II Indices | Abbr. | Description | Unit | Weight |
|---|---|---|---|---|---|---|
| | | GDP per capita | B1 | Indication of economic level | CNY | 0.0628 |
| | | Population | B2 | Indication of population distribution | $1 \times 10^4$ people | 0.0452 |
| | Driver | Urbanization | B3 | Indication of the structure of urban and rural residents | % | 0.0613 |
| | | Natural population growth | B4 | Indication of the population growth | % | 0.0203 |
| | | Large industrial energy consumption | B5 | Indication of the resource consumption | $1 \times 10^4$ t | 0.1396 |
| | Pressures | Agricultural fertilizer application rate | B6 | Indication of the environmental pollution | t | 0.0516 |
| | | Atmospheric SO$_2$ concentration | B7 | Indication of the environmental pollution | ug/m$^3$ | 0.0396 |
| | Status | Domestic water consumption | B8 | Indication of residents' living standard | $1 \times 10^8$ m$^3$ | 0.0594 |
| Ecological risk indicator system | | Per capita disposable income | B9 | Indication of the living standard | CNY | 0.0484 |
| | | Forest cover rate | B10 | Quality of the ecological environment | % | 0.0348 |
| | | Shannon diversity index | B11 | Indication of landscape diversity and heterogeneity | - | 0.0376 |
| | Impacts | Plaque area variation coefficient | B12 | Indication of the impact of plaque changes on ecosystems | - | 0.0444 |
| | | Agglomeration index | B13 | Indication of the impact on the ecosystem from human activity | - | 0.0485 |
| | | Plaque density | B14 | Indication of the fragmentation of the landscape | Count/km$^2$ | 0.0547 |
| | | Tertiary industry proportion | B15 | Indication of the industrial structure | % | 0.0594 |
| | Responses | Cultivated area | B16 | Indication of the regional land structure | $1 \times 10^4$ Mu | 0.0482 |
| | | Total water resources | B17 | Indication of the regional policy response and environmental protection techniques | $1 \times 10^8$ m$^3$ | 0.0738 |
| | | Rainfall | B18 | Indication of the quality of the ecological environment | mm | 0.0705 |

(2)  Calculation of the comprehensive ecological risk index

Ecological risks are the effects of potential accidents and disasters on the ecosystem or its components, which may damage the structure and function of the ecosystem and therefore threaten the safety of the ecosystem. The risks to the ecosystem come from various sources. In order to accurately and comprehensively assess the ecological risk, the ERI in Chongqing was assessed based on the DPSIR frameworks in this study. Details of the calculation are shown as follows:

$$ERI = \sum_{c=1}^{n} R_c \times a_c \tag{3}$$

where $R_C$ is the weight of index $C$, and $a_C$ is the normalized value of index $C$.

### 2.3.4. Development of the ES-DPSIR Model

The coordinated development of ecosystem services and ecological risks is highly associated with the well-being of humans. In order to evaluate the coordination between them, a double index-based ES-DPSIR model was developed in this study. The principle of this model is to assess the degree of coordination through dispersion, where a larger dispersion indicates a lower degree of coordination. The coordination is calculated as follows:

$$EC = 2 \times \left[ \left( \frac{ESI \times ERI}{ESI + ERI} \right)^{-2} \right]^{1/2} \tag{4}$$

where $EC$ is the coordination degree (from 0 to 1). Higher values of $EC$ indicate a relatively high coordination between ecosystem services and ecological security. The coordination was divided into 5 levels using the equidistant method, including dissonance (0.0–0.2), mild dissonance (0.2–0.4), mild coordination (0.4–0.6), moderate coordination (0.6–0.8), and high coordination (0.8–1.0).

The evolution trends of ecosystem services and ecological security are characterized by the coordinated development index:

$$ED = (EC \times ET)^{1/2} \tag{5}$$

$$ET = a \times ESI + b \times ESI \tag{6}$$

where $ED$ is the coordinated development degree, and $ET$ is the weighted $ESI$ and $ERI$. In this study, 0.5 was used for both $a$ and $b$, since ecosystem services and ecological risks are equally important to ecosystems. The coordination development degree was divided into 5 levels using the equidistant method as well, including highly coordinated (0.8–1.0), moderately coordinated (0.6–0.8), mildly coordinated (0.4–0.6), mildly dysregulated (0.2–0.4), and dysregulated (0.0–0.2).

## 3. Results

### 3.1. Analysis of the Comprehensive Ecosystem Service Index (ESI)

The results of the ecosystem service indicator in Chongqing are shown in Figure 3. The distribution of different ecosystem service indicators showed significant spatial differences. Low grain production areas ($<1.1 \times 10^5$ t) were mostly found in the main urban area due to the high urbanization and the dominance of construction in land use. The highest grain production was found in Kaizhou, Wanzhou, Hechuan, Jiangjin, and Yongchuan, with production ranging from 4.44 to $6.91 \times 10^5$ t. The low water supply areas were mainly in Northeastern Chongqing (197–660 mm), whereas high water supply was mainly found in the southeastern and western regions of Chongqing (up to 1564 mm). In 2020, the soil conservation service in Chongqing was moderate, with an overall value of between 0 and $1.15 \times 10^3$ t, indicating a stable soil conservation service function. The highest soil conservation service function was found in some areas in the northeastern and southeastern regions of Chongqing. Significantly higher values of air purification services were found in the southeastern and northeastern regions of Chongqing, which were closely associated with the distribution of forests. The lush vegetation coverage the southeastern and northeastern regions of Chongqing could assist with air purification. The overall carbon storage in Chongqing was relatively high, with more than 6 t in most areas, indicating a good carbon sequestration ecosystem in Chongqing. Similar spatial distributions were found between climate regulation and habitat quality, with a lower score near the main urban area due to the higher population density and more frequent human activity in that area. Higher tourist populations were found in Wanzhou, Yunyang, and the main urban area and surroundings, which was mainly caused by the locally developed tourism resources and

frequent economic activity. In contrast, lower tourist populations were found in Chengkou, Wuxi, and Xiushan due to the inconvenient transportation and underdeveloped economies. The results of the spatial differences in the evaluation of different ecosystem service indicators show that anthropogenic factors were a significant cause of the spatial differences in each ecosystem service and that human disturbances in areas with a dense distribution of the population were intense and had caused serious damage to surface ecosystem elements, resulting in a low level of ecosystem service function. In addition, the combined effects of natural factors such as precipitation, temperature, and geomorphologic patterns influenced the distribution of ecosystem services. The main urban area of Chongqing, for example, had the largest population distribution and density, a large urbanized construction area, and strong impacts on ecosystems from anthropogenic disturbances, which resulted in a generally low level of ecosystem services in the region. On the other hand, areas such as southeast and northeast Chongqing had relatively low population density, and the combined effects of high precipitation, strong evapotranspiration, and topography resulted in a pattern of outstanding ecosystem services in the region.

The results of the ESI are shown in Figure 4. In general, the lowest ESI was found in the main urban area, with increasing scores the further out the region was. The ESI in the main urban areas (i.e., Beibei, Shapingba, Yuzhong, Jiulongpo, Yubei, Bishan, and Nan'an) ranged from 0.13 to 0.181. The low scores were mainly caused by serious restrictive effects on the environment from high urbanization and frequent human activity, which limited the capability of ecosystem services. A clear improvement in the ESI was found in the surrounding districts and counties of the main urban area (i.e., Dazu, Yongchuan, Ba'nan, Chaoshou, Liangping, etc.), ranging from 0.182 to 0.268. Further improvement was found in Hechuan, Qijiang, Nanchuan, Fulin, Fengdu, Shizhu, Kaizhou, Chengkou, Qianjiang, and Xiushan, ranging from 0.269 to 0.445. These areas covered 26.3% of the districts and counties in Chongqing, indicating that urban human activity gradually declined with increased distance from the main urban area. The highest ESI scores (0.446–0.689) were found in the outermost areas, which were least affected by the main urban area and had the highest level of ecosystem services. These areas included Jiangjin, Wulong, Pengshui, Youyang, Kaizhou, Yuyang, Fengjie, Wuxi, and Wushan, covering 23.7% of the districts of Chongqing. Overall, the ESI scores and spatial trends suggested that the capacity of ecosystem services to function well was associated with the distance from urban economic activity, showing a concentric outward radiation pattern.

### 3.2. Analysis of the Comprehensive Ecological Risk Index (ERI)

Figure 5 shows the results of the DPSIR in Chongqing in 2020, where significant spatial trends of indicators were also observed. Per capita GDP, urbanization, per capita disposable income, and domestic water consumption showed similar spatial trends, with the highest level in the main urban area and decreasing trends the further out the region was. This was mostly caused by the frequent economic activity in the main urban area. This economic activity also led to the lower forest coverage rate and cultivated land area in the main urban area, as well as the agricultural fertilizer application rate, as the usage of fertilizer was highly associated with the cultivated land area. Construction dominated land types in the main urban area, whereas cultivated land was distributed in other areas. Higher natural population growth rates were found in the southeastern and northeastern regions of Chongqing, which were associated with the local economy and population. Atmospheric concentrations of $SO_2$ were relatively stable across different districts and counties, possibly due to the airflow in the study area. The Shannon diversity index, plaque area variation coefficient, agglomeration index, and plaque density were heavily affected by land use, surface landscape, and human activity. For example, more plaque fragmentations led to a higher plaque area variation coefficient and agglomeration index. Aquatic resources and rainfall also showed a high similarity in spatial trends, because the northeastern and southeastern regions of Chongqing received more humid and hot air from the southeast, resulting in more precipitation.

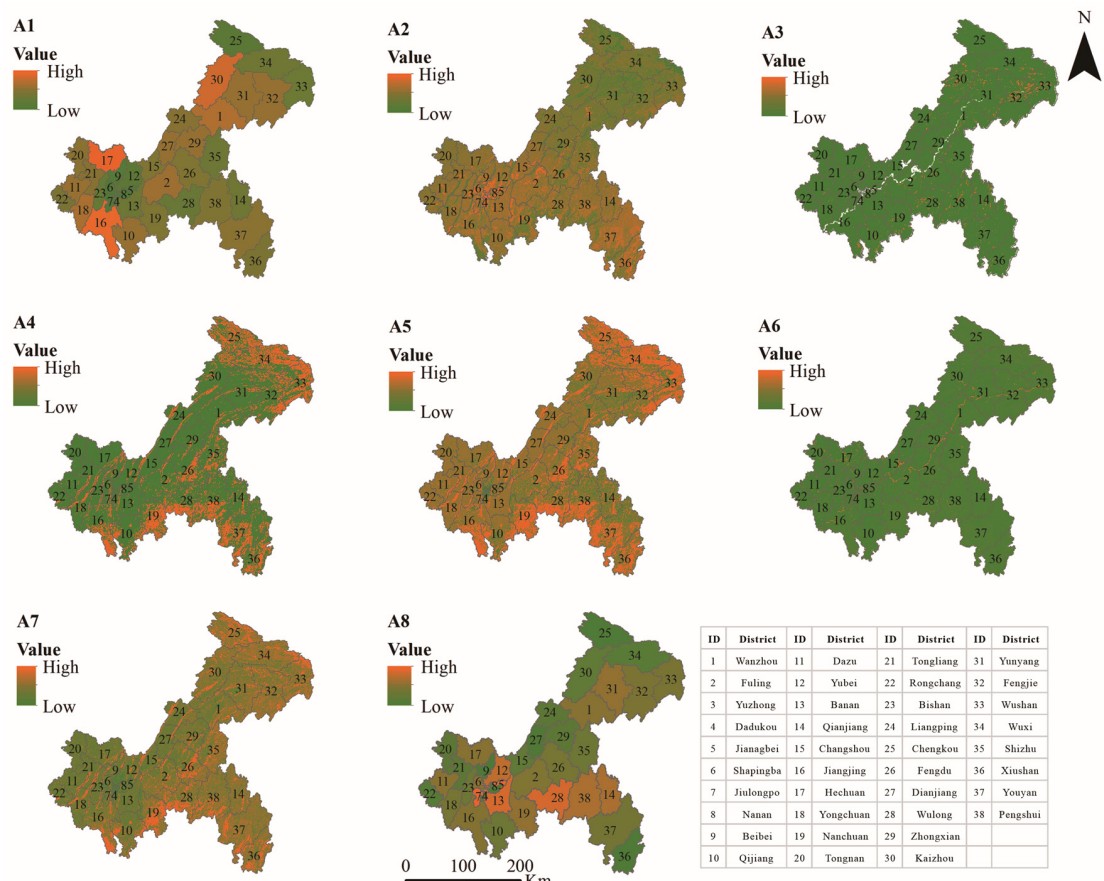

The following table appears within the figure:

| ID | District | ID | District | ID | District | ID | District |
|----|----------|----|----------|----|----------|----|----------|
| 1 | Wanzhou | 11 | Dazu | 21 | Tongliang | 31 | Yunyang |
| 2 | Fuling | 12 | Yubei | 22 | Rongchang | 32 | Fengjie |
| 3 | Yuzhong | 13 | Banan | 23 | Bishan | 33 | Wushan |
| 4 | Dadukou | 14 | Qianjiang | 24 | Liangping | 34 | Wuxi |
| 5 | Jiangbei | 15 | Changshou | 25 | Chengkou | 35 | Shizhu |
| 6 | Shapingba | 16 | Jiangjing | 26 | Fengdu | 36 | Xiushan |
| 7 | Jiulongpo | 17 | Hechuan | 27 | Dianjiang | 37 | Youyan |
| 8 | Nanan | 18 | Yongchuan | 28 | Wulong | 38 | Pengshui |
| 9 | Beibei | 19 | Nanchuan | 29 | Zhongxian | | |
| 10 | Qijiang | 20 | Tongnan | 30 | Kaizhou | | |

**Figure 3.** Results of the ecosystem service index in Chongqing in 2020. (A1: Food production; A2: Water supply; A3: Soil conservation; A4: Air purification; A5: Carbon storage; A6: Habitat quality; A7: Climate regulation; A8: Leisure).

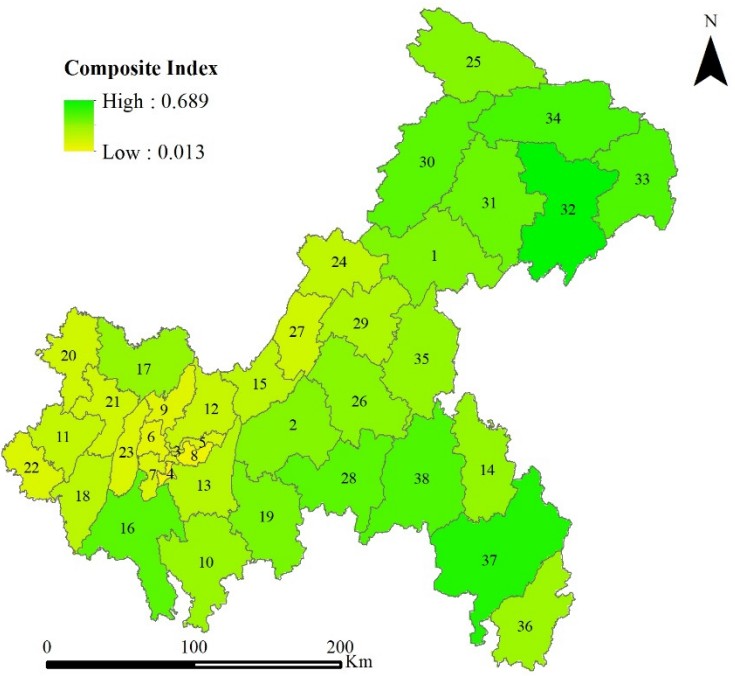

**Figure 4.** Results of the comprehensive ecosystem service index.



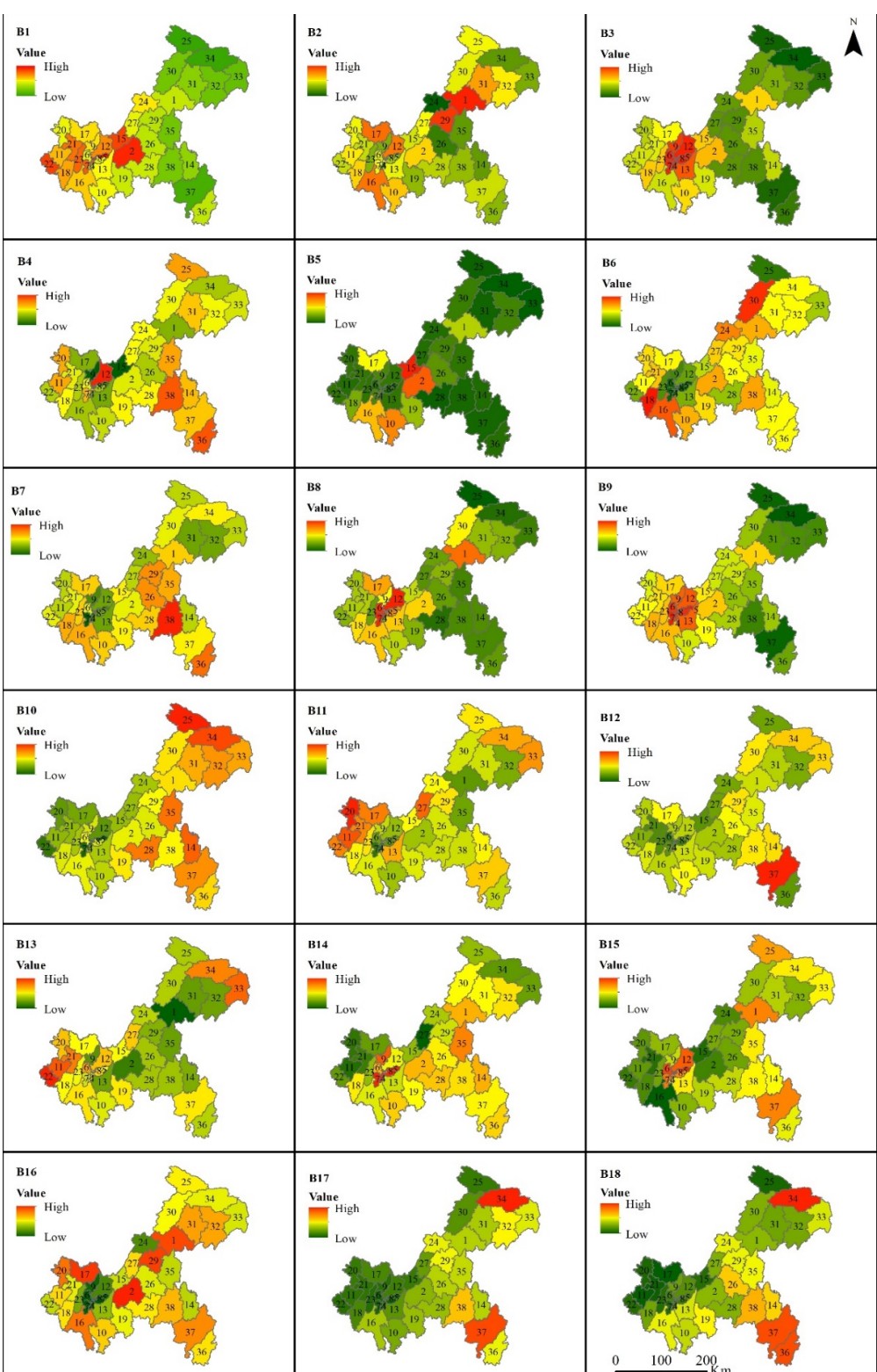

**Figure 5.** Visualization results of the DPSIR model. (B1: GDP per capita; B2: Population; B3: Urbanization; B4: Natural population growth; B5: Large industrial energy consumption; B6: Agricultural fertilizer application rate; B7: Atmospheric $SO_2$ concentration; B8: Domestic water consumption; B9: Per capita disposable income; B10: Forest cover rate; B11: Shannon diversity index; B12: Plaque area variation coefficient; B13: Agglomeration index; B14: Plaque density; B15: Tertiary industry proportion; B16: Cultivated area; B17: Total water resources; B18: Rainfall).

The results of the comprehensive ERI are shown in Figure 6, where it can be seen that the spatial trends were more random compared with those of the ESI. The lowest ERI values were found in Fuling, Fengdu, Pengshui, Shizhu, Wanzhou, and Fengjie, ranging from −0.143 to −0.039, indicating the lowest ecological risks in those areas. The higher vegetation coverage, cultivated area, water supply, and development of the ecological environment could explain the ecological safety in such areas. The lowest ESI scores were found in Wuxi, Wushan, Yuyang, Youyang, Changshou, and Dazu, ranging from −0.040 to −0.086. Slightly higher ESI scores (0.087–0.144) were found in nine districts (23.7% of total districts and counties, including Tongnan, Hechuan, Tongliang, Yubei, etc.), and further increased scores (0.145–0.236) were found in some areas in the western and northeastern regions of Chongqing (i.e., Yongchuan, Jiangjin, Ba'nan, Beipei, Kaizhou, and Chengkou), whereas the highest scores (0.237 to 0.333) were found in Rongchang, Bishan, Nanchuan, Dianjiang, Liangping, and Qianjiang. Overall, the northeastern and southeastern regions of Chongqing had much lower ERI scores, indicating compromised ecological safety in those areas. This was because of the fragmentation of the surface landscape, decreased vegetation cover caused by economic development, and urbanization in the main urban areas and western Chongqing, which eventually increased the local ecological risks.

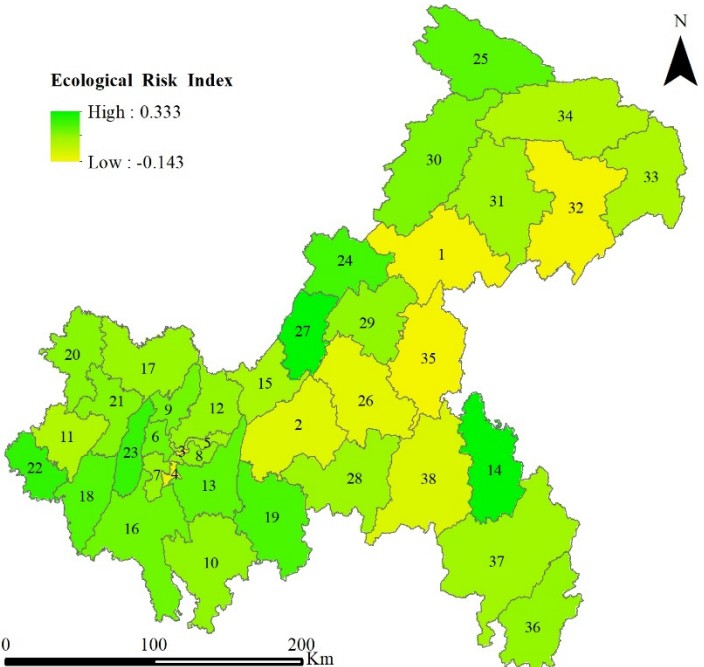

**Figure 6.** Results of the comprehensive ecological risk indices (ERI).

### 3.3. Analysis of the ES-DPSIR Model

The coordination between ecosystem services and ecological risks in Chongqing showed an increasing trend from west to east (Figure 7, left panel). For example, imbalanced areas were found in the west (i.e., Rongchang, Tongnan, Beipei, and Bishan) and the main urban areas, accounting for 31.58% of all of the districts and counties. The coordination of Dazu, Yongchuan, Yubei, Jiangbei, Ba'nan, and other districts and counties increased and was in a state of mild imbalance. In terms of spatial distribution, ecosystem services–ecological risks were mainly distributed at the junction of the western, northeastern, and southeastern regions of Chongqing, accounting for 52.63% of the total districts and counties. By combining the results of ecosystem service assessment (Figure 4) and ecological risk assessment (Figure 6), it was determined that the low ecosystem service index and the high ecological risk in these areas were significant reasons for the imbalance. The districts and counties with the highest coordination degree were Pengshui, Fengjie, and Wuxi, which were in a highly coordinated state. As the results of Figures 4 and 6 show, the composite

index of ecosystem services and ecological risk in these areas was at a high level, indicating that these areas had a high contribution to ecosystem services but at the same time had potentially large ecological risk. On the whole, the spatial relationship between ecosystem services and ecological risks in Chongqing's districts and counties had great contradictions, and further management and regulation of the relationship between the two needs to be carried out.

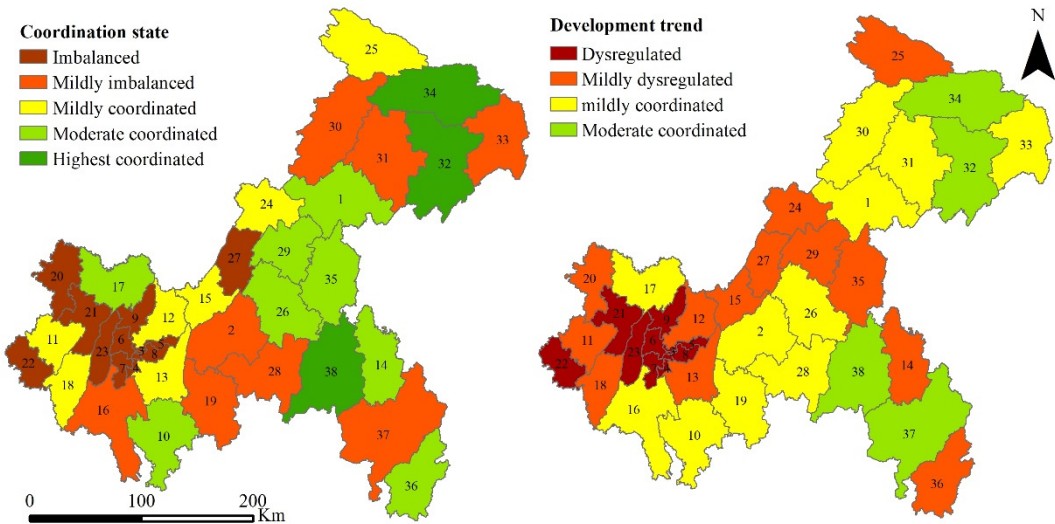

**Figure 7.** Spatial relationship and development trend of ecosystem services and ecological security in Chongqing.

The evolving trends of the spatial relationships between ecosystem services and ecological risks showed significant differences in different areas (Figure 7, right panel). In total, 60.53% of the districts and counties showed mildly dysregulated and dysregulated development. Dysregulated development was found mostly in the main urban areas and in West Chongqing (i.e., Rongchang, Tongliang, Bishan, Jiangbei, Shapingbai, Jiulongpo, and Nan'an), whereas slightly dysregulated development was found in Tongnan, Dazu, Yongchuan, Yubei, Banan, Chagnshou, Dianjiang, Liangping, Zhongxian, Shizhu, Qianjiang, and Chengkou. Dysregulated development was mainly due to the low contribution of ecosystem services and the increase in ecological risks. In addition, 39.49% of the districts showed mildly and moderately coordinated development, including Hechuang, Jiangjin, Qijiang, Nanchuan, Fulin, Wulong, Fengdu, Wanzhoum Kaizhaou, Yunyang, and Wushan with mildly coordinated development, and Pengshui, Youyang, Fengjie, and Wuxi with moderately coordinated development. The reason for coordinated development was the increased contribution of ecosystem services, but at the same time, it had greater potential ecological risks. Therefore, more attention should be paid to the imbalanced and mildly imbalanced developed areas in the future to coordinate the improvement of ecological services and decrease ecological risks. Overall, the contradiction between the spatial relationship between ecosystem services and ecological risks in some areas of Chongqing was still prominent, and it is necessary to further improve ecosystem services, reduce ecological risks, and optimize future development contradictions.

## 4. Discussion

### 4.1. Advantages of the ES-DPSIR Model in Analyzing the Spatial Relationships of Ecosystem Services and Ecological Risks

Ecosystem services and ecological risks are related to the well-being of humans [41,42]. Ecosystem services can provide continuous and stable products and services for the development and daily lives of humans [43], whereas ecological risk control can ensure the stability of our habitat [44]. A few studies have quantitatively evaluated the ecological services and ecological risks, with a focus on one or the other. There are very limited studies

on the relationship between ecological services and ecological risks. However, these studies only assess the overall relationship between them, which is not effective in improving the balance between ecological services and risks or understanding their driving factors. In addition, existing studies lack an analysis of the relationship between the two from the morphology of spatial change, and the results make it difficult to promote the coordinated optimization of ecosystem services and ecological security.

In this study, a double index-based ES-DPSIR model was developed to assess the relationships between ecosystem services and ecological risks on the basis of spatial patterns and to demonstrate the spatial relationship status and evolution of trends between these systems. The results for the statuses indicate the levels of spatial relationship development between ecological services and ecological risks, whereas the evolution predicts developments in the near future that could help us better understand the relationship between ecological services and risks and therefore implement relevant policies to improve ecological safety.

### 4.2. Analysis of the Driving Factors of the Spatial Relationship Development of Ecosystem Services and Ecological Risks

There are certain associations and interactions between factors in the ecosystem [12,45]. Therefore, the spatial development of ecosystem services and ecological security is affected by complex ecological and environmental factors [46–49]. To optimize the spatial relationship development pattern of ecosystem services and ecological risks, the factor analysis model in SPSS 19.0 was used to analyze the driving factors of different indices in the ES-DPSIR model. Based on the principles of factor analysis, principal components (PC) with eigenvalues greater than 1 were selected for further analyses [48], with an accumulated contribution of 84.71% from the first six components. The score for each principal component is shown in Table 4. The score of most components (except for PC 4 and 5) was above 0, including food supply, water supply, soil conservation, air purification, carbon storage, habitat quality, climate regulation, and tourism. The positive scores from such indicators suggest that they could have promoted the spatial relationships that increased ecosystem services and reduced ecological risks. For example, in PC 1, the scores of water supply, air purification, carbon storage, and habitat quality were greater than 0.15, indicating that these factors were positively associated with vegetation landscape structure and negatively associated with local ecological risks [49–51]. However, scores of indicators for different PCs varied in the DPSIR model, indicating the uncertainties of the driving factors. Among the scores of the indices in the DPSIR model, per capita GDP, natural population growth rate, atmospheric $SO_2$ concentration, and plaque density showed negative scores for most of the PCs, with minimum scores of $-0.250$, $-0.497$, $-0.145$, and $-0.323$, respectively, indicating an increase in regional ecological risk and a reduction in the function of ecosystem services. Per capita GDP and natural population growth rate represented the human interference with ecosystem services, especially in areas with high economic development and rapid population growth. This was mainly because of the decreased ecosystem service capacity and increased ecological risks, which were caused by plaque fragmentation and environmental pollution due to the excessive development of ecosystems [38,52]. Natural indicators such as vegetation coverage, cultivated land, water resources, and rainfall showed positive scores in most PCs, indicating that those factors could have promoted the improvement in ecosystem services and ecological security in most cases, despite some restraints to a certain degree. The variation in scores may have been caused by the interaction between indicators [38]. For example, excessive rainfall could result in soil erosion, debris flow, and other geological disasters, which could decrease the vegetation coverage, damage the soil structure, worsen the surface fragmentation, and eventually not be conducive to the enhancement of ecosystem services and the stable improvement of regional ecological security.

**Table 4.** Principal component loading scores.

| Indices | Principal Component 1 | Principal Component 2 | Principal Component 3 | Principal Component 4 | Principal Component 5 | Principal Component 6 |
|---|---|---|---|---|---|---|
| A1 | 0.061 | 0.064 | 0.079 | 0.110 | −0.027 | 0.011 |
| A2 | 0.246 | 0.029 | 0.059 | −0.278 | −0.026 | 0.063 |
| A3 | 0.130 | 0.033 | 0.060 | −0.022 | 0.015 | 0.020 |
| A4 | 0.150 | 0.017 | 0.063 | −0.038 | −0.020 | 0.029 |
| A5 | 0.186 | 0.110 | 0.069 | −0.140 | −0.049 | 0.197 |
| A6 | 0.156 | 0.080 | 0.064 | −0.023 | 0.024 | 0.045 |
| A7 | 0.041 | 0.187 | 0.030 | 0.061 | 0.050 | 0.068 |
| A8 | 0.057 | 0.099 | 0.381 | 0.037 | 0.058 | 0.029 |
| B1 | −0.045 | −0.004 | 0.150 | −0.096 | 0.098 | −0.250 |
| B2 | 0.003 | 0.350 | 0.148 | −0.143 | 0.006 | −0.160 |
| B3 | −0.065 | 0.022 | 0.089 | 0.040 | −0.120 | 0.042 |
| B4 | −0.092 | 0.105 | −0.046 | 0.027 | −0.043 | −0.497 |
| B5 | 0.016 | 0.053 | −0.145 | 0.016 | −0.136 | 0.476 |
| B6 | −0.097 | 0.115 | −0.064 | 0.200 | 0.006 | 0.035 |
| B7 | −0.145 | −0.012 | −0.084 | 0.380 | −0.034 | 0.076 |
| B8 | −0.093 | 0.241 | 0.198 | 0.037 | −0.103 | −0.169 |
| B9 | −0.104 | 0.052 | 0.078 | 0.071 | −0.077 | 0.046 |
| B10 | 0.185 | 0.060 | −0.054 | 0.189 | −0.035 | −0.013 |
| B11 | −0.031 | 0.013 | 0.024 | −0.022 | 0.359 | −0.074 |
| B12 | −0.055 | 0.043 | 0.126 | 0.287 | 0.004 | −0.112 |
| B13 | −0.009 | −0.086 | 0.173 | 0.048 | 0.301 | 0.046 |
| B14 | −0.020 | −0.076 | −0.028 | 0.107 | −0.323 | 0.060 |
| B15 | 0.050 | −0.027 | 0.320 | 0.048 | −0.001 | −0.105 |
| B16 | 0.020 | 0.187 | 0.002 | −0.020 | 0.053 | 0.008 |
| B17 | 0.012 | −0.054 | 0.086 | 0.255 | −0.004 | −0.005 |
| B18 | −0.091 | −0.123 | 0.059 | 0.464 | −0.063 | 0.004 |

In summary, ecological service indicators such as water supply, air purification, etc., were the driving factors for maintaining regional ecological security. In addition, economic and social indicators such as per capita GDP and natural population growth rate were the main reasons for the decline in ecosystem services and the increase in ecological risks, which could have disrupted the spatial contradiction of ecosystem services and ecological risk. Several natural indicators, including rainfall, could have generally promoted the level of ecological risk in the region, but in some circumstances these indicators could also have worsened the contradiction between them. Therefore, to improve the spatial relationship development of ecosystem services and ecological risks, it is necessary to improve ecosystem services such as food production and air purification (especially water supply and soil conservation), which affect rainfall, vegetation cover, and habitat quality [50]. At the same time, it is also important to adjust human activity, reduce landscape fragmentation, reduce pollution-related problems caused by land use and sewage discharge, and improve local biological safety.

### 4.3. Suggestions and Prospects

There must be some relationships between regional ecosystem services and ecological risks that are difficult to express directly [51,53]. In practice, the improvement of regional ecosystem services can promote the reduction of regional ecological risks and increase the pattern of regional ecological security [23]. However, affected by natural, man-made, and other comprehensive factors, places with high ecosystem services do not necessarily have a high ecological security index and may still face greater ecological risks [38]. Therefore, it is of great significance to explore the relationship between ecosystem services and ecology. In this research, based on the innovation system of the dual index framework of ecosystem services and ecological risks, the coupled coordinated analysis model was introduced to explore the spatial relationship between ecosystem services and ecological risks at county

scales and reflect their spatial change trends. The results show that the relationship pattern and future change trend can be explained in space, which can provide methods and path references for related research.

Although the research explains the spatial relationship between ecosystem services and ecological risks well and explores the control drivers of drivers on ecosystem service–ecological risk synergies, the impact on overall ecosystem services was not considered in the research. One of the reasons is that ecosystems are complex processes, and in order to accurately demonstrate mechanisms, more comprehensive influencing factors need to be considered. In addition, ecosystem service–ecological risk change is a continuous process of spatiotemporal dynamics, and a study of the change mechanism should pay attention to the analysis of spatiotemporal sequences. In future research, attention should be paid to considering more influencing factors and spatiotemporal sequences.

## 5. Conclusions

This study identified and analyzed the spatial relationship and evolutionary trend of ecosystem services and ecological risks in Chongqing at the district and county scales by innovatively constructing the ES-DPSIR model, which provides new ideas for the study of the relationship. The main conclusions are as follows: (1) The lowest comprehensive ESI was found in the main urban areas of Chongqing, with improvement increasing the further out a region was. The ESI values in the main urban areas, city surroundings, and outermost areas ranged from 0.13 to 0.181, from 0.182 to 0.268, and from 0.446 to 0.689, respectively. (2) The spatial distribution of the ERI was more random than the ESI. The ERI values ranged from $-0.146$ to $-0.039$ in Fuling, Fengdu, etc.; from $-0.040$ to $-0.086$ in Wuxi, Wushan, etc.; from 0.087 to 0.144 in Tongnan, Hechuan, etc.; from 0.145 to 0.236 in Yongchuang, Jiangjin, etc.; and from 0.237 to 0.333 in Rongchagn, Bishan etc. (3) The current situation of the ecosystem services and ecological risk spatial relationship pattern was prominently different from east to west. A total of 52.63% of the districts and counties in Chongqing had an imbalanced or mildly imbalanced status. (4) The evolution trends of the spatial relationships between ecosystem services and ecological risks showed significant differences. In total, 60.53% of the districts and counties showed imbalanced and mildly imbalanced development, and the increase in ecological risks in the future was the main cause of the imbalance development.

At present, there is a strong international focus on the need for ecosystems to provide a steady stream of benefits to humans, a requirement that is based on the need to protect or restore damaged ecosystems. However, while pursuing their own development, human beings are bound to cause ecosystem damage, either intentionally or unintentionally, which inevitably leads to an increase in ecological risk, resulting in a contradictory relationship between ecosystem services and ecological risk. Current research often ignores the relationship between ecosystem services and ecological risk, and this study focused on the relationship between ecosystem services and ecological risk by constructing an ES-DPSIR model to investigate the spatial relationship and trends of the two. It is of great significance to identify the driving indicators that lead to ecological risk or reduced ecosystem services through the analysis of driving mechanisms and then reduce ecological risk to improve ecosystem services and improve the contradictory relationship between the two. The study explains the spatial relationship between ecosystem services and ecological risk well and discusses the driving factors of the two. However, the study did not consider the comprehensive impact of ecosystem services, mainly because the interaction of environmental variables in ecosystems is complex and cannot be accurately grasped. In addition, based on the temporal and dynamic nature of ecosystem evolution, future research should pay attention to considering more influencing factors and spatial and temporal sequences.

**Author Contributions:** Methodology, Z.L. and K.Z.; software, Y.B.; formal analysis, H.W.; investigation, D.G.; resources, Y.Z. (Yanjun Zhang); data curation, J.C. and X.S.; writing—original draft, Z.L.; writing—review and editing, K.Z.; funding acquisition, D.S. and Y.Z. (Ya Zhang). All authors have read and agreed to the published version of the manuscript.

**Funding:** The research reported in this manuscript was funded by the Scientific Research Project of Chongqing Ecological Environment Bureau (No. CQEE2022-STHBZZ118), the National Natural Science Foundation of China (42171298), the Natural Science Foundation of Chongqing in China (cstc2020jcyj-jqX0004), the General Program of Natural Science Foundation of Chongqing Science and Technology Bureau: Construction of Risk Measurement Model for Agricultural Non-point Source Pollution in Mountain Areas and Analysis of Future Development Scenarios (CSTB2022NSCQ-MSX0538), and the Comprehensive Survey on Ecological Restoration in Alpine Meadow Area of Northwest Yunnan, China (DD20230482).

**Institutional Review Board Statement:** Not applicable.

**Informed Consent Statement:** Not applicable.

**Data Availability Statement:** Due to privacy or ethical constraints, the data could not be accessed through the creation.

**Conflicts of Interest:** The authors declare no conflict of interest.

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
