# Peer review of "Analysis of Spatial Relationship Based on Ecosystem Services and Ecological Risk Index in the Counties of Chongqing"

_land, doi:10.3390/land12101830_

Round 1
Reviewer 1 Report
This paper is very interesting but has the following problems: the specifics of ecosystem services are unclear. It is necessary to explain which areas and to what extent ecosystem services differ and whether these differences are due to anthropogenic or natural factors.
Author Response
This paper is very interesting but has the following problems: the specifics of ecosystem services are unclear. lt is necessary to explain which areas and to what extent ecosystem services differ and whether these differences are due to anthropogenic or natural factors.
Response:Thanks for your comments and suggestions. We have carefully considered and incorporated the views of the reviewer and added a description of the extent of differences in ecosystem services in section 3.1, with further discussion and analysis of the factors contributing to the differences. For more details, please refer to the corresponding section 3.1 in the revised manuscript.
Reviewer 2 Report
Please see attachment

Quality of English writing (grammar) is good.
Author Response
1.The manuscript does not contain Line number. The authors shall use the prescribed MDPI Land template.
Response:Thanks for your comments and suggestions. We have reformatted the formatting as required by MDPI.
2.In Section 2.1 (Study Area), include the coordinated of the municipality, and a figureshowing the location of the study area that contains the topographic contours(topographic map).
Response:Thanks for your advice. In response to the reviewer's suggestion, we have added a description of the location of the study area and a schematic diagram of the location of the study area. For more details, please refer to the corresponding section in the revised manuscript.
3.It would be helpful if Figure 1 includes full description. Also, it would be best if this figure is placed near the paragraph where it has been cited.
Response:Thanks for your comments and suggestions. We have carefully considered and adopted the opinions of experts, and have re-described the framework contents of the original Figure 1 in full detail and added relevant references. For details, please refer to the corresponding section in the revised manuscript.
4.Data used in this study were secondary data. Authors shall include evidence/s on how they manage in quality assurance of the data.
Response:Thanks for your comments and suggestions. We have carefully considered and adopted the reviewers' suggestions, and we have re-described the study data and added evidence on quality assurance for management data.
5.It would be better if Tables and Figures under Section 2 shall be placed close to the paragraph where it has been cited.
Response:Thanks for your comments and suggestions. We have reformatted the document according to the formatting requirements of the MDPI and placed the relevant charts and graphs below the corresponding paragraphs. For details, please refer to the corresponding section in the revised manuscript.
6.All acronyms shall be defined during its first use in the manuscript such as the ES-DPSIR.
Response:Thanks for your comments and suggestions. We have defined all acronyms as they should be used for the first time in a manuscript, based on expert advice. For more details, please refer to the corresponding section in the revised manuscript.
7.The ES-DPSIR concept is not the first for this paper, hence, needs citation.
Response:Thanks for your comments and suggestions. We have added citations to the corresponding places in the manuscript in accordance with the reviewers' comments. For more details, please refer to the corresponding section in the revised manuscript.
8.Section 2.3.2 page 2 shall describe how the indicators were selected.
Response:Thanks for your comments and suggestions. We have carefully considered and incorporated the reviewers' suggestions, and we have added a rationale and description of the selection of the indicators in section 2.32, explaining in further detail why these indicators were selected for the study. For details, please refer back to the original revised draft.
9.Consideration of entropy weight’needs supporting evidences or citations.
Response:Thanks for your comments and suggestions. We have added supporting evidence and citations at the corresponding entropy rights in the manuscript based on the experts' suggestions. For more details, please refer to the corresponding section in the revised manuscript.
10.Include a map on soil types, flood hazard maps, and other related hazards. These are all contributory to ecological risks.
Response:Thanks for your comments and suggestions. We have carefully considered the recommendations of the experts, soil type and flood disaster are the elements that cause ecological risk, the ecological risk in this study is mainly concerned with the ecological environment state by the combined effect of nature and society, according to the DPSIR model from the natural ecological environment and the social environment aspects of the selection of indicators, of which the soil type in the calculation of the indicator of the soil conservation service has already been taken into account, and the flood disaster and the precipitation in the study area has a close relationship, these indicators have been included in the mapping of the indicators.
11.Soil quality shall be clearly describe especially on metals and metalloids concentration as these are ecological risks.
Response:Thanks for your comments and suggestions. We have carefully considered the suggestions given by the reviewers. Ecological risk is concerned with the state of the ecological environment, which is a combination of nature and society, and this paper selects indicators based on the DPSIR model from both the natural ecological environment and the social environment, highlighting the ecological risk from a macro perspective, and does not explore the ecological risk caused by soil quality from a micro perspective, but this paper selects the amount of agricultural fertilizer application and the landscape pattern index that can laterally reflect the quality of the soil.
12.Needs necessary citations in Section 2 and inclusion of more references in the List of References
Response:Thanks for your comments and suggestions. We have added relevant citation and more references based on expert advice. For more details, please refer to the corresponding section in the revised manuscript.
Reviewer 3 Report
It is an interesting paper for assess a region or a county. Although, I think, a classification map of the region could be useful to the reader understand the forcings (drivers) that operate in the region. Others physiographic features like altimetry, slope, etc could hep the reader to better understand the region to be analysed.
Author Response
It is an interesting paper for assess a region or a county. Although, I think, a classification map of the region could be useful to the reader understand the forcings (drivers) that operate in the region. Others physiographic features like altimetry, slope, etc could hep the reader to better understand the region to be analysed.
Response:Thanks for your comments and suggestions. We have added a map of the slope of the study area and the spatial distribution of ecosystem types at the location of the study area, based on the opinions of the reviewer. In addition, we have adjusted the colors in the map to help the reader better understand the area to be analyzed. For more details, please refer to the corresponding section in the revised manuscript.
Round 2
Reviewer 2 Report
Excellent work.